# Pragmatic multicentre randomised controlled trial evaluating the impact of a routine molecular point-of-care 'test-and-treat' strategy for influenza in adults hospitalised with acute respiratory illness (FluPOC): trial protocol

Kate Beard,[1] Nathan Brendish,[1,2] Ahalya Malachira,[2] Samuel Mills,[2] Cathleen Chan,[2] Stephen Poole,[1,3] Tristan Clark[1,2,3]

For numbered affiliations see end of article.

**Correspondence to**
Dr Kate Beard;
kate.beard@uhs.nhs.uk

## ABSTRACT

**Background** Influenza infections often remain undiagnosed in patients admitted to hospital due to lack of routine testing. When tested for, the diagnosis and treatment of influenza are often delayed due to the slow turnaround times of centralised laboratory PCR testing. Newer molecular systems, have comparable accuracy to laboratory PCR testing, and can generate a result in under 1 hour, making them potentially deployable as point-of-care tests (POCTs). High-quality evidence for the impact of routine POCT for influenza on clinical outcomes is, however, currently lacking. This large pragmatic multicentre randomised controlled trial aims to address this evidence gap.

**Methods and analysis** The FluPOC trial is a pragmatic, multicentre, randomised controlled trial evaluating adults admitted to a large teaching hospital and a district general hospital with an acute respiratory illness, during influenza season and defined by Public Health England. Up to 840 patients will be recruited over up to three influenza seasons, and randomised (1:1) to receive either POCT using the FilmArray respiratory panel, or routine clinical care. Clinical and infection control teams will be informed of the results in real time and where influenza is detected clinical teams will be encouraged to offer neuraminidase inhibitor (NAI) treatment in accordance with national guidelines. Those allocated to standard clinical care will have a swab taken for later analysis to allow assessment of missed diagnoses. The outcomes assessment will be by retrospective case note analysis. The outcome measures include the proportion of influenza-positive patients detected and appropriately treated with NAIs, isolation facility use, antibiotic use, length of hospital stay, complications and mortality.

**Ethics and dissemination** Prior to commencing the study, approval was obtained from the South Central Hampshire A Ethics Committee (reference 17/SC/0368, granted 7 September 2017). Results generated from this protocol will be published in peer-reviewed scientific journals and presented at national and international conferences.

**Trial registration number** ISRCTN17197293

### Strengths and limitations of this study

► This is a multicentre randomised controlled trial.
► The pragmatic trial design more accurately reflects routine clinical care in a hospital setting.
► This is a highly relevant clinical question currently given the variable use of influenza testing and neuraminidase inhibitors in clinical practice, the ongoing threat of further influenza pandemics, the recent development of novel influenza antivirals and the undefined role of influenza point-of-care testing in acute respiratory illness during seasonal influenza.
► The inclusion, of some patients lacking capacity and older patients often with multiple comorbidities, makes the trial population generalisable to a secondary care population.
► There is ongoing uncertainty however as to how the suggested 'test-and-treat' strategy could be best implemented in clinical practice.

## BACKGROUND

### Respiratory virus burden of disease

Respiratory tract infections are the second most common cause of mortality and morbidity worldwide,[1] with viruses the most frequently detected pathogens in adults hospitalised with acute respiratory illness.[2] Seasonal influenza epidemics lead to excess hospitalisations and death due to complications including pneumonia and exacerbation of underlying cardiopulmonary

conditions, and occur mainly in the elderly and patients with comorbidity.[3–5]

### Influenza: widespread but underdiagnosed

Estimates of the burden of influenza virus infection in hospitalised adults have traditionally been based on the incidence of the influenza-like-illness syndrome (ILI, defined as fever of >38°C and new respiratory symptoms) which has poor sensitivity (around 50%) and specificity (0%–63%), rather than on laboratory-confirmed influenza.[6–9] Patients may also present as decompensated cardiovascular disease, collapse or diabetic emergencies.[10 11] A recent Canadian study estimated that only around 1 in 14 emergency department (ED) visits due to influenza virus infection was correctly attributed to influenza.[12]

Sampling for respiratory viruses is generally performed on upper respiratory tract samples; however, several recent studies suggest that in lower respiratory tract syndromes (such as pneumonia and exacerbation of chronic obstructive pulmonary disease (COPD)) testing of upper respiratory tract samples for viruses is insensitive compared with testing lower respiratory tract samples.[13 14]

For these reasons, it is likely that the burden of influenza and other respiratory viruses among hospitalised adults and its economic impact have been vastly underestimated.

### Neuraminidase inhibitors

Neuraminidase inhibitors (NAIs) such as oseltamivir (Tamiflu) are recommended by Public Health England (PHE) and WHO guidelines for all hospitalised adults with suspected and proven influenza infection.[15] Although there have been no randomised controlled trial evaluating the efficacy of NAIs in this group, well-controlled observational data suggest that the treatment of influenza with NAIs reduces mortality in hospitalised adults, especially when commenced rapidly.[16 17] Therefore, it is important that hospitalised patients with influenza are all identified and treated as soon as possible after presentation.

### Previous point-of-care tests (POCTs) for influenza

Rapid diagnostic tests for influenza, based on antigen detection in upper respiratory tract samples, have been available for many years but have poor diagnostic accuracy in adults, where sensitivity is around 50%.[18 19] The poor sensitivity of these antigen-based tests has limited their clinical utility and their use was not associated with clinical or health economic benefits in a large randomised controlled trial in hospitalised adults.[20] The current gold standard diagnostic test for respiratory viruses is laboratory-performed PCR which is highly sensitive and specific but has a typical turnaround time for results of 24–48 hours.[21] New rapid, molecular tests have recently been developed, including the BioFire FilmArray respiratory panel. These molecular platforms are broadly equivalent in accuracy to laboratory PCR, without the need for specialist laboratory support and expertise, can provide a result in under 1 hour and can potentially be used as a POCT.

### FilmArray respiratory panel

The FilmArray respiratory panel 2 (BioFire Diagnostics, Utah, USA) uses nested real-time PCR to detect around 20 respiratory viruses. The FilmArray requires only 2 min of 'hands on' time and produces a test result in around 45 min.[22] The FilmArray respiratory panel 2 is both Food and Drug Administration (FDA) cleared and CE in-vitro diagnostic (IVD) marked. The viral pathogens detected by the FilmArray respiratory panel 2 include: influenza A (H1 and H3), influenza B, adenovirus, coronaviruses (HKU1, NL63, 229E, OC43) human metapneumovirus, human rhinovirus/enterovirus, parainfluenza (types 1–4) and respiratory syncytial virus.[20 21] The FilmArray respiratory panel is broadly equivalent in accuracy to laboratory PCR, and has been validated on nose and throat swabs, nasopharyngeal aspirates, lower respiratory tract samples and on samples from immunocompromised patients.[23–30] These studies also show high reliability, ease of use and short turnaround times.

### Clinical impact of POCT for respiratory viruses

Though viruses are the most commonly detectable pathogen in adults hospitalised with acute respiratory illness, antibiotic use is near universal.[2] Antibiotic overuse for acute respiratory illness is partly driven by clinical uncertainty regarding the underlying aetiology; therefore, early identification of viruses may prevent or shorten unnecessary antibiotic use. Previous small studies have suggested that rapid molecular viral testing might reduce antibiotic use and improve influenza antiviral use.[31 32] We have recently shown in a large pragmatic randomised controlled trial (ResPOC)[33] that routine syndromic molecular POCT for respiratory viruses (using the FilmArray respiratory panel) in adults presenting to hospital with acute respiratory illness was associated with a number of clinical benefits compared with routine clinical care including a reduction in unnecessary antibiotics use, and a reduced length of hospital stay. It also suggested improved influenza detection and antiviral use and better side room utilisation, although the number of patients infected with influenza was small and the study was not powered to evaluate outcomes in this subgroup.[33] This follow-on study seeks to definitively evaluate the impact of routine molecular POCT for influenza on the detection of patients infected with influenza, their clinical care and outcome. Accepting the unpredictability of influenza activity year on year, this study is designed to recruit larger numbers of patients infected with influenza, and powered to evaluate the impact of POCT on national guideline recommended antiviral use. In addition, the novel use of sampling and deferred viral testing in the control group allows a direct assessment of missed diagnosis of influenza in hospitalised adults.

### Alignment with global research priorities

In addition to being focused on patient and healthcare organisation outcomes, this clinical research is strongly

aligned with several global research priority initiatives including WHO's Battle against Respiratory Viruses (BRaVe) initiative[34] and the UK national report into antibiotic resistance.[35] The BRaVe initiative aims to catalyse multidisciplinary research on strategies to prevent and treat medically important respiratory virus infections with the goal of timely integration of research advances into public health practice. Priority areas identified include improving diagnostic tests for viral respiratory illness and improving the clinical management of patients with acute respiratory viral illness, both addressed by this study. With the growing threat of antimicrobial resistance to global public heath, there is increasing need for the research and development of interventions which can help avoid unnecessary antibiotics use.[36] The data that will be generated by this trial regarding the impact of influenza POCT on antibiotic prescribing is highly aligned with this global initiative.

## Added value

In addition to evaluating the clinical benefit of routine molecular POCT for influenza, we will also use this opportunity to answer other important research questions including the ideal specimens for respiratory virus testing (upper vs lower respiratory tract specimens) in patients with lower respiratory tract infection (LRTI), the difference in viral load between these two specimen type and whether changes in viral load over time (kinetics) in respiratory samples or the presence of viral RNA in the blood is related to and predictive of clinical outcome.

## STUDY AIMS AND OBJECTIVES
### Aims

This study aims to prospectively evaluate the impact of a routine molecular point-of-care (POC) 'test-and-treat' strategy for influenza in adults hospitalised with acute respiratory illness during influenza season.

### Objectives

1. To evaluate the impact of routine molecular POCT on the clinical management of influenza including: the detection rate of influenza, the adherence to national (PHE) influenza guidelines, the speed and appropriateness of NAI use and the speed and appropriateness of isolation facility use.
2. To evaluate the impact of routine molecular POCT on measures of clinical effectiveness in adults hospitalised with influenza including: time to clinical stability, time on supplementary oxygen, duration of hospitalisation, adverse events and mortality.
3. To explore the differences in viral detection between upper and lower respiratory tract samples and the changes in viral load that occur over time in NAI-treated and NAI-untreated patients.

Study processes that patient–participant will undergo are summarised in figure 1.

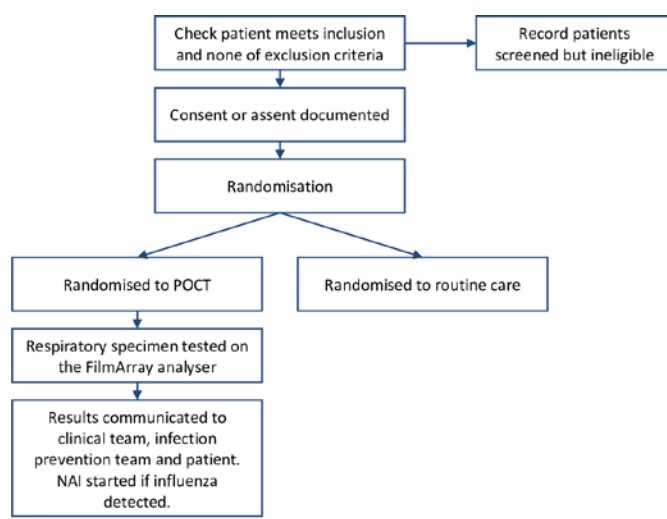

**Figure 1** Participant flow through the study. NAI, neuraminidase inhibitor; NIHR WTCRF, National Institute for Health Research WellcomeTrust Clinical Research Facility; POCT, point-of-care test.

## METHODS
### Trial design

This is a pragmatic, multicentre, parallel group, open-label, randomised controlled study. Groups are allocated 1:1 to the intervention arm (POCT) and control arm (routine clinical care). This protocol adheres to the Standard Protocol Items: Recommendations for Interventional Trials statement.[37] The study will take place across up to three influenza seasons (as defined by national PHE surveillance programmes).

## Participants, interventions and outcomes
### Study setting

This is a multicentre study based within secondary care at University Hospital Southampton (UHS) National Health Service (NHS) Foundation Trust, Southampton, UK, and Royal Hampshire County Hospital, Hampshire Hospitals NHS Foundation Trust, Winchester, UK. Patients will be recruited from the acute medical unit (AMU) and the ED at these hospital sites.

### Eligibility criteria
#### Inclusion criteria

► Is a patient in an ED* or AMU, within UHS NHS Foundation Trust or Hampshire Hospitals NHS Foundation Trust.
► Aged ≥18 years old.
► Has acute respiratory illness**.
► Duration of respiratory illness less than 10 days prior to hospitalisation.

► Can be recruited to the study within 16 hours of presentation.

*For patients in ED, a decision will have already been made that the patient will be admitted.

**An episode of acute respiratory illness is defined as an acute pulmonary illness (including pneumonia, bronchitis and influenza-like illness) or an acute exacerbation of a chronic respiratory illness (including exacerbation of COPD, asthma or bronchiectasis). For the study, acute respiratory illness as a provisional, working, differential or confirmed diagnosis must be made by a treating clinician.

### Exclusion criteria

► Not fulfilling all the inclusion criteria.
► A purely palliative approach being taken by the treating clinicians.
► Previously included in this study and re-presenting within the last 30 days after hospital discharge.
► Declines nasal/pharyngeal swabbing.
► Consent declined or consultee consent declined.

Concurrent, prior or subsequent enrolment in an observational study is not necessarily an exclusion criterion; this is at the discretion of the chief investigator and will be assessed on a case-by-case basis.

### Interventions

#### For both control and intervention groups

A respiratory sample will be collected by a member of research staff. A combined nose and throat swab will be taken and placed into viral transport medium. If feasible, a lower respiratory tract sample (sputum) will also be obtained.

Clinical and demographic data will be collected at the time of enrolment. Outcome data will be collected from case notes and electric health records retrospectively.

A blood sample (maximum of 21 mL) will be obtained from all willing participants, processed in the National Institute for Health Research (NIHR) Wellcome Trust Clinical Research Facility lab and stored at −80°C pending further analysis. All respiratory samples leftover from FilmArray testing may be frozen and stored for further study. Participant consent for this is included in the consent form. Patients may be approached for additional samples to be collected and stored for further study (see next for schedule of additional samples).

A letter detailing that the patient has been included in this trial will be sent to the patient's general practitioner for information only.

#### For those randomised to the interventional arm

The nose and throat swab in viral transport medium will be analysed promptly on the FilmArray respiratory panel 2 as per training delivered by the apparatus manufacturer, used as a POCT. FilmArray machines will be located in the clinical areas for this purpose. Test results are generated in around 45 min. Where lower respiratory tract samples are also obtained, these will also be tested on the FilmArray in parallel with the nose and throat swab or sequentially.

The results of the test will be documented in the patient's case notes and a member of the clinical team responsible for the patient will be directly informed. The participant or consultee will also be informed of the result where appropriate. The infection prevention and control team will be informed of the result within 4 hours of testing. In the event of influenza A or B being detected, clinical teams will be encouraged to offer treatment with NAIs as per national guidelines. It is known that the sensitivity of virus detection may be greater with lower respiratory tract samples versus upper tract samples in certain patients. Therefore, in patient where lower respiratory tract samples are obtained in addition to the upper respiratory tract samples, if either of the swabs are positive they will treated as infected and be reported to the clinical team and infection control team as above.

#### For those randomised to the control group

These patients will be managed according to standard clinical care as directed by their treating clinicians. Laboratory respiratory virus testing will be at the discretion of the responsible clinical team and where performed will be using laboratory PCR (in the onsite laboratory). The respiratory samples collected at enrolment will be stored at −80°C and subsequently tested by the FilmArray respiratory panel at least 30 days after collection. This will allow direct comparison of pathogens between the groups (ie, an estimate of missed diagnoses in the routine clinical care group) but will not influence participant care.

### Additional procedures

For participants randomised to receive the intervention, who subsequently have a pathogen detected on the FilmArray respiratory panel and for all participants randomised to routine clinical care, additional respiratory samples ±an additional blood test (maximum 10 mL) may be specifically requested from the participants to study temporal changes in viral load, according to the schedule described next.

#### Schedule

The nose and throat swab and lower respiratory tract samples listed here are in addition to samples taken at enrolment (day 1) and are collected at five further time points: days 2–3, days 4–5, days 6–7, days 8–10 and days 11–14+. An additional blood sample (maximum of 10 mL) may be taken at days 5–7. The participant may decline all or any of these additional tests.

### Primary outcome

The primary outcome measure is the difference in the proportion of influenza-positive patients treated appropriately with NAI during their hospital stay (within 5 days of admission).

### Secondary outcomes

Proportion of cases of influenza identified.

Proportion of all NAI use occurring in influenza-positive patients.

Proportion of all NAI use occurring in influenza-negative patients.

Median time from admission to NAI commencement, hours.

Median duration of NAI use in influenza-positive patients, days and doses.

Median duration of NAI use in influenza-negative patients, days and doses.

Proportion of patients treated with antibiotics.

Proportion of patients treated with single doses or brief courses (<24 hours) of antibiotics.

Median duration of antibiotic use, days.

Proportion of patients isolated in a side room.

Median duration of isolation facility use.

Proportion of influenza cases correctly isolated.

Median time from admission to isolation of influenza-positive cases.

Median time from admission to deisolation of influenza-negative cases.

Median duration of hospitalisation, days.

Median time to clinical stability*, days.

Median time on supplementary oxygen, days.

Proportion of patients with intensive care unit (ICU) or high dependency unit (HDU) admission.

Median duration of ICU or HDU stay, days.

Proportion re-presenting to hospital within 30 days.

Proportion readmitted to hospital within 30 days.

Proportion of in hospital, 30-day and 60-day mortality.

Median turnaround time for viral testing.

All outcomes are measured for the duration of hospitalisation or up to 30 days from recruitment/admission (whichever is shortest) unless specified otherwise and include medication (antibiotics and NAIs) that patients are discharged home with.

*Defined as: temperature ≤37.8°C, respiratory rate ≤24 breaths/min, heart rate ≤100 beats/min, oxygen saturation ≥90% without the use of supplementary oxygen, systolic blood pressure ≥90 and normal mental status mm Hg for at least 24 hours.[38]

### Exploratory outcomes

Difference in influenza detection between upper and lower respiratory tract samples.

Difference in influenza viral load between upper and lower respiratory tract samples.

Proportion with influenza RNA detected in blood.

Changes in influenza viral load over time (kinetics) in respiratory tract samples and blood.

Response rate on the ordinal hospital recovery scale at day 4 and/or 7.

### Participant timeline

Participants are identified in the AMU and ED according to eligibility criteria by research staff. Following gaining written consent, participants are immediately randomised to the intervention group or the control group. For those patients randomised to the intervention group, a nose/throat swab is collected by a study doctor or research nurse and this is tested on the FilmArray machine. If the patient is able to produce a sputum sample, this sample will also be tested on the FilmArray machine. Results are generated in approximately 45 min and are immediately communicated to the clinical team. Clinical data are collected prospectively and retrospectively for both groups. Further nose/throat swabs, sputum samples and blood tests may be collected from patients who remain inpatients on serial follow-up visits up to day 14.

### Sample size
#### Proposed sample size
The study will recruit up to 840 patient–participants: about 280 per season for three consecutive influenza seasons. With 1:1 allocation to groups, there will be 420 patients per group. On the conservative assumption of a 25% positivity rate for influenza during influenza season (ie, ~100 influenza-positive patients in each group), the group sizes would give a 90% power at a 0.05 significance level to detect a difference of 20% in appropriate NAI use (from 65% to 85%) and an 80% power to detect a difference of 17%. In our previous work, 91% of influenza-positive patients randomised to POCT were appropriately treated with NAI versus 65% with routine clinical care, a difference of 26% (95% CI 10 to 43). We suspect that the difference may be larger in this study as there will likely be a number of undiagnosed (and therefore untreated) cases in the control group revealed by testing them with the FilmArray at a later date. In the event of higher numbers of influenza-positive patients being recruited in the first two seasons, the study may be stopped prematurely (once ~100 influenza-positive patients have been recruited to the intervention group).

### Recruitment and screening
#### Screening
Eligible patients in the AMU and the ED will be identified by research staff via regular review of the comprehensive admissions information technology (IT) systems daily. Recruitment will run for up to 3 years to include three seasonal influenza seasons between December 2017 and May 2020.

#### Consent
The study team (research nurse, research fellow or the principal investigator) will obtain informed consent for those fulfilling eligibility criteria and willing to be recruited for those with capacity, or assent via consultee for those without capacity, as per the dedicated study forms. In view of the acute nature of patients' illnesses, the potential benefits of rapid identification of viruses including prompt NAI use for influenza, the usual 24 hours consideration period for a participant or consultee, will not apply.

Discussion of the study will be provided to patients, or their consultee for those lacking capacity, by study staff.

This includes supply of a participant information sheet for the participant or witness to read and retain. Consent is also acquired to obtain and store specimens (nose/throat swab, sputum sample, blood tests) for future research studies.

Each patient will be assumed to have capacity unless it is established that they lack capacity. For patients unable to consent for themselves, this study complies with the Mental Capacity Act 2005 and in such cases, the patient's family member, carer or friend may be asked to act as the personal consultee and provide assent. In the event of a personal consultee not being available, a nominated consultee (usually the consultant caring for the patient and independent from the study) will be asked if they would provide assent. Both the person taking assent and the consultee must personally sign and date the relevant form.

### Assignment of interventions
#### Sequence generation, allocation concealment and implementation
Following screening and obtaining informed written consent from an eligible patient, a unique identification number will be assigned consecutively. A research team member will then use this identification number to randomise the participant to either the intervention or control group, via an internet-based randomisation service (sealedenvelope.com). This service uses permuted blocks of varying sizes and generates a randomisation code which corresponds to one of the study arms. Research staff will use the allocation code generated from sealedenvelope.com to assign the patients to the intervention or control group.

#### Blinding
As this is a pragmatic trial of a diagnostic strategy where the results of the test must be communicated to clinical and infection control teams, no attempt at blinding trial participants, research staff or care providers will be made. Data analysts will be blinded to group allocation.

### Data collection, management and analysis
#### Data collection methods
Demographic and clinical data will be collected for all patients at enrolment from paper and electronic medical records including: age, sex, ethnicity, smoking status, vaccination status, comorbidities, medication use, symptoms, duration of illness prior to hospitalisation, observations (pulse rate, respiratory rate, blood pressure, oxygenation status), laboratory results, radiology results, antimicrobial and antiviral use prior to hospitalisation and provisional diagnosis. Data will be recorded on a standardised case report form (CRF) and transferred to a secure electronic database. Once patients have been discharged or after 30 days (whichever is soonest), outcome data will be collected retrospectively from electronic and physical case notes, electronic prescribing systems and laboratory and radiological results systems including: use of antivirals, duration of antivirals, time from assessment to antiviral use, use of antibiotics, duration of antibiotic use, use of side room facilities, time from assessment to isolation facility use, time to clinical stability, duration of supplementary oxygen use, duration of hospitalisation, complications including ICU and HDU admission, representation and readmission to hospital within 30 days, final diagnosis and mortality. The number of diagnostic tests and procedures performed will also be recorded and data will also be collected on the turnaround time of respiratory virus test results in each group. Patients withdrawn from the study will have no further data collected.

#### Data management
The study will be conducted in compliance with the approved protocol, relevant International Conference on Harmonisation (ICH) and Good Clinical Practice (GCP) regulations and standard operating procedures. Data will be evaluated for protocol compliance and accuracy in reference to the source documents.

The subjects' anonymity will be maintained. The study team will keep a log of each subject's name, hospital identification (ID) number, date of birth and unique participant trial number. The participant details will be recorded on the secure NHS Edge system in a similar manner, including NHS number. This participant trial number is used on documents after screening to maintain confidentiality. Documents that are not anonymous (eg, signed informed consent forms) will be maintained separately, in strict confidence and in accordance with the Data Protection Act 1998. A secure database will be used to enter data from the CRFs at the completion of the study, followed by data lock.

The study staff will be responsible for entering study data in the CRF. It is the investigators' responsibility to ensure the accuracy of the data entered in the CRF.

Only the research study team will know the identity of subjects and have access to the list linking participant details to the participant trial number.

### Statistical methods
Analysis will be by intention to treat (ITT). The framework is superiority. Trial results will be reported in accordance with the Consolidated Standards of Reporting Trials (CONSORT) statement.[39]

Statistical analysis will be performed by a dedicated medical statistician from the University of Southampton, independent from the study team. No interim analysis is currently planned. Patients tested with the rapid molecular diagnostic test will be compared with patients managed according to standard clinical care using standard descriptive and comparative statistical methods using Prism (GraphPad Software, La Jolla, California), and Stata (StataCorp).

Missing data were minimal (<2%) in the CI's previous POCT study involving 720 patients presenting to secondary care, and is not expected to be significant higher in this trial; however, the use of multiple imputation will be performed should missing data exceed

5% for the primary outcome or key secondary outcome measures.[40]

Summaries of all baseline characteristics will be presented using means and SD, medians and IQRs, or frequencies and percentages, as appropriate. The intervention and control groups will be compared using ORs and differences in proportions for equality of proportions for binary data (eg, proportion of cases of influenza detected) and using independent-sample t-tests or non-parametric equivalent as appropriate for continuous data (eg, duration of antibiotics, test turnaround time, etc).

### Primary outcome

The primary outcome (proportion of patients infected with influenza treated with NAIs) will be measured only in patients with confirmed influenza, that is, the intention-to-treat-infected (ITTI) population. Difference in proportions and unadjusted ORs will be used to compare the groups. The effect of group on the primary outcome will be further assessed using multiple logistic regression to control for the following covariates: age, sex, influenza vaccine status, receipt of antibiotics prior to admission, duration of illness, comorbidity, temperature, C reactive protein, severity of illness and clinical group.

Certain secondary outcome measures will be measured both in the entire cohort (ITT population) and in the influenza-infected cohort (the ITTI population) while others will only be measured in the ITTI population, as detailed in the outcome section. The intervention and control groups will be compared as per the primary outcome for equality of proportions for binary data and using t-tests and non-parametric equivalent tests for continuous data (eg, turnaround time) as appropriate.

In view of the clinical heterogeneity of patients infected with influenza, heterogeneity of effect among different clinical subgroups is anticipated a priori and therefore a preplanned subgroup analysis for key secondary outcome measures may be undertaken based on clinical group (exacerbation of asthma, exacerbation of COPD, pneumonia, ILI, non-pneumonic LRTI and 'other') and also based on receipt or otherwise of antiviral treatment (the receipt of NAIs within 5 days of admission). The interaction between clinical subgroups and allocation group will be assessed in the multiple regression models detailed above.

### Monitoring

This trial was reviewed by the sponsor. On the basis of the very low risk of harms associated with the intervention in this non-Clinical Trial of an Investigational Medicinal Product (CTIMP) trial, no data monitoring committee or interim analysis is planned. A trial management committee will meet regularly and oversee the trial.

### Harms

The risks of respiratory tract sampling and additional blood tests being taken are minimal and where occurring are likely to be mild. No additional adverse events related

to POCT for respiratory viruses are anticipated. However, active monitoring and reporting of severe adverse events will be undertaken. A serious adverse event (SAE) is any adverse event that:

► Results in death
► Is life threatening.
► Requires hospitalisation or prolongation of existing hospitalisation.
► Results in persistent or significant disability or incapacity.
► Consists of a congenital anomaly or birth defect.

Participants already admitted to AMU or are in ED but with a decision already made to admit are considered already hospitalised. However, an adverse event leading to prolongation of their existing hospitalisation will be counted as an SAE.

### Auditing

Regular monitoring by the sponsor will be performed according to ICH GCP. Data will be evaluated with regards to accuracy in relation to source documents and compliance with the protocol. Following written standard operating procedures, the monitors will verify that the clinical trial is conducted and data are generated, documented and reported in a manner that is compliant with the protocol, GCP and applicable regulatory requirements.

### Protocol amendments

There has been one amendment made to the protocol. This was the addition of Hampshire Hospitals NHS Foundation Trust as a potential site and removal of mention of 'single centre'. The current protocol version is V.1.1, dated 23 November 2017. This amendment has been communicated to the investigators and the trial registries. The local study reference is RHM MED 1438.

### Confidentiality

All data will be anonymised to ensure patient confidentiality is protected. A unique study number will be used to identify participant data in the CRFs and database. SAEs will be reported in line with GCP and regulatory requirements. All study staff have been trained in GCP. Data will be kept securely and only the investigators and sponsor's representative (monitor) have access to the data.

### Access to data

Full access to the dataset will be held by the principal investigator, coinvestigators and independent statistician only. The dataset may be made available to other parties on request.

### Dissemination policy

Authorship of this manuscript and those resulting from this protocol will follow the International Committee of Medical Journal Editors (ICMJE) recommendations and CONSORT statement where appropriate, and there are no plans to use professional writers. There is no intention to make the dataset publically available. Outside of the study team and regulators, the full protocol will only be

made available at the discretion of the chief investigator. Multiple publications are expected to result from the data and samples collected, and these publications must acknowledge this trial and the study team as appropriate.

## Patient and public involvement

Discussion with members of the public and the results of questionnaire from hospitalised patients helped inform the design of this trial.

The emphasis on informing the patients, in addition to the clinical teams, of the POCT result is a direct result of our engagement with patients. Patients and the public will also be involved in the reporting of this research.

## DISCUSSION

We hypothesise that the use of a routine molecular POC 'test-and-treat' strategy for influenza in adults hospitalised with acute respiratory illness will improve adherence to national guidelines for the management of influenza and may improve patient outcomes. We expect increased, timely detection of influenza and that this will lead to improved use of NAI treatment. Rapid and appropriate use of NAI treatment in influenza may also lead to decreased complication and mortality and may shorten length of stay in hospital. The rapid detection of viruses may improve isolation facility use within the NHS with subsequent reduction in nosocomial transmission of viruses and improvements in patient flow through acute areas. The use of a POCT for respiratory viruses may also identify patients where unnecessary antibiotic use can be safely discontinued, promoting antibiotic stewardship and reducing antimicrobial resistance.

**Author affiliations**
[1]Academic Unit of Clinical and Experimental Sciences, Faculty of Medicine, University of Southampton, Southampton, UK
[2]Department of Infection, University Hospital Southampton NHS Foundation Trust, Southampton, UK
[3]NIHR Southampton Biomedical Research Centre, University Hospital Southampton NHS Foundation Trust, Southampton, UK

**Acknowledgements** The authors would like to thank patients and staff in the emergency department and acute medical unit and other inpatient wards at University Hospital Southampton NHS Foundation Trust and Hampshire Hospitals NHS Foundation Trust for making the study possible.

**Contributors** TC conceived the study and is the chief and principal investigator. TC and NB designed the study. TC and NB wrote the protocol. KB wrote the protocol manuscript. AM, SM, CC and SP reviewed the manuscript. All authors read and approved the final manuscript.

**Funding** The study is funded by NIHR Postdoctoral Fellowship Programme, grant number: PDF-2016-09-061. The testing units and equipment for this study will be leased or purchased independently from a UK distributor using funds provided by NIHR Post Doctoral Fellowship Programme (PDF-2016-09-061). NIHR Clinical Research Network, Wessex, provided salary contributions for clinical fellows to support this trial. The study is sponsored by University Hospital Southampton NHS Foundation Trust.

**Competing interests** None declared.

**Patient consent for publication** Not required.

**Ethics approval** Prior to commencing the study, approval was obtained from the South Central Hampshire A Ethics Committee (reference 17/SC/0368). A protocol

amendment has been approved by the ethics committee for the addition of Hampshire Hospitals NHS Foundation Trust as a potential site and the removal of mention of 'single centre' (current protocol version V.1.1, date 23 November 2017). The local study reference number is RHM MED 1438.

**Provenance and peer review** Not commissioned; externally peer reviewed.

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
