## [Reviewer comments · BMJ Open]

ARTICLE DETAILS

TITLE (PROVISIONAL)	Pragmatic multicentre randomised controlled trial evaluating the impact of a routine molecular Point-of-Care 'test-and-treat' strategy for influenza in adults hospitalised with acute respiratory illness (FluPOC): trial protocol
AUTHORS	Beard, Kate; Brendish, Nathan; Malachira, Ahalya; Mills, Samuel; Chan, Cathleen; Poole, Stephen; Clark, Tristan

VERSION 1 – REVIEW

REVIEWER	Ben Cooper University of Oxford, UK Mahidol Oxford Tropical Medicine Research Unit, Thailand
REVIEW RETURNED	13-Jun-2019

GENERAL COMMENTS	I have just a handful of suggestions for improvements (mostly to improve clarity and remove ambiguity) which can be made at the authors' discretion: 1. In the background section it would be helpful to explicitly spell out the distinction between this study and the earlier large RCT mentioned on p122. p22 line 47 Once patients have been discharged or after 30 days .” 30 days from what date?3. p25, line 32 “The use of multiple imputation will be considered should missing data exceed 10% for the primary outcome.” There are no hard and fast rules, but 10% cut-off for considering imputation methods seems quite high and 5% has been suggested as a rule of thumb elsewhere https://bmcmedresmethodol.biomedcentral.com/articles/10.1186/s12874-017-0442-14. Also, it would better if the decision rules for using multiple imputation (MI) methods could be spelled in some detail (eg. as in figure 1 in the above link) rather than just saying “MI will be considered”.5. p 24 line 53 "The effect of group on the primary outcome” Presumably clinical group is meant here. Can these clinical groups be specified?6. p26 , line 19. “pre-planned subgroup analysis...” . How will subgroups be defined in terms of age, clinical group and NAI treatment? i.e what will the categories be?7. No mention is made of costs or resource use implications of the rapid test, but these are important components needed for any cost-effectiveness analysis which will be important for informing decisions about using the test. Worth at least considering collecting some such data as part of the study.
--

REVIEWER	Dr Nasir Wabe Macquarie University, Australia
REVIEW RETURNED	07-Aug-2019

GENERAL COMMENTS	Manuscript Title: Pragmatic multicentre randomised controlled trial evaluating the impact of a routine molecular Point-of-Care 'test-and-treat' strategy for influenza in adults hospitalised with acute respiratory illness (FluPOC): trial protocol Comments to Authors: This is a study protocol aimed to prospectively evaluate the impact of a routine molecular Point-of-Care 'test and treat' strategy for influenza in adults hospitalised with acute respiratory illness. The protocol is generally well written and clearly presented. The protocol would benefit if the following information is added/clarified.  • Please move "Trial design" -page 13 to 'Methods' section (preferably after 'Study setting'). • Please add strength and limitation in the discussion section. • The study design is a 'randomised controlled study' and page 20 (screening) describes the recruitment process as....."Recruitment will run for up to 3 years to include 3 seasonal influenza seasons between December 2017 and May 2020". Is retrospective or prospective design. Please clarify?
---

VERSION 1 – AUTHOR RESPONSE

Reviewer: 1

Reviewer Name: Ben Cooper

Institution and Country: University of Oxford, UK; Mahidol Oxford Tropical Medicine Research Unit, Thailand Please state any competing interests or state 'None declared': None declared

Please leave your comments for the authors below I have just a handful of suggestions for improvements (mostly to improve clarity and remove ambiguity) which can be made at the authors' discretion:

1. In the background section it would be helpful to explicitly spell out the distinction between this study and the earlier large RCT mentioned on p12

Author's response

Many thanks for this comment. We have now added a paragraph to the background section to clarify this - page 13 .

2. p22 line 47 Once patients have been discharged or after 30 days ." 30 days from what date?

Author's response

Many thanks for this comment. This is 30 days from recruitment (and also 30 days from admission as participants must be recruited within 16 hours of admission) we have now clarified this in the text at various points.

3. p25, line 32 "The use of multiple imputation will be considered should missing data exceed 10% for the primary outcome."

There are no hard and fast rules, but 10% cut-off for considering imputation methods seems quite high and 5% has been suggested as a rule of thumb elsewhere

<https://eur03.safelinks.protection.outlook.com/?url=https%3A%2F%2Fbmcmcdresmethodol.biomedcentral.com%2Farticles%2F10.1186%2Fs12874-017-0442-1&data=01%7C01%7CT.W.Clark%40soton.ac.uk%7C8b980c5ef89c4554656108d726eff3aa%7C4a5378f929f44d3ebe89669d03ada9d8%7C0&odata=W1hga%2FZ5HWons8E8fD6OtnvkDxHwi4SSiEcZEESI13A%3D&reserved=0>

Author's response

Many thanks for this helpful comment. After further discussion with the trial statistician we agree that setting the cut off for missing data at 10% is higher than standard according to many and so have altered this to 5% as suggested. We have added a new reference to support this.

4. Also, it would better if the decision rules for using multiple imputation (MI) methods could be spelled in some detail (eg. as in figure 1 in the above link) rather than just saying "MI will be considered".

Author's response

Many thanks for this comments. We agree and have now added this to the methods section.

5. p 24 line 53 "The effect of group on the primary outcome" Presumably clinical group is meant here. Can these clinical groups be specified?

Author's response

Many thanks for this comment. The 'effect of group on the primary outcome' refers to the allocation group (POCT or control), analysed further by adjusting for covariates (that might influence the decision to use NAIs) using multivariate analysis.

6. p26 , line 19. "pre-planned subgroup analysis..." . How will subgroups be defined in terms of age, clinical group and NAI treatment? i.e what will the categories be?

Author's response

Many thanks for this comments. We have now clarified the outcomes that will be evaluated in subgroup analysis and have limited the subgroups to clinical groups (which is now defined in the text) and NAI treatment (also now defined in the text).

7. No mention is made of costs or resource use implications of the rapid test, but these are important components needed for any cost-effectiveness analysis which will be important for informing decisions about using the test. Worth at least considering collecting some such data as part of the study.

Author's response

Many thanks for this comments. So we are indeed collecting health resource data in this study as detailed in the data collection section. The health economic analysis will be a subsequent additional study with a separate protocol.

Reviewer: 2

Reviewer Name: Dr Nasir Wabe

Institution and Country: Macquarie University, Australia Please state any competing interests or state 'None declared': None declared

Please leave your comments for the authors below Manuscript Title:

Pragmatic multicentre randomised controlled trial evaluating the impact of a routine molecular Point-of-Care 'test-and-treat' strategy for influenza in adults hospitalised with acute respiratory illness (FluPOC): trial protocol Comments to Authors:

This is a study protocol aimed to prospectively evaluate the impact of a routine molecular Point-of-Care 'test and treat' strategy for influenza in adults hospitalised with acute respiratory illness. The protocol is generally well written and clearly presented. The protocol would benefit if the following information is added/clarified.

- Please move "Trial design" -page 13 to 'Methods' section (preferably after 'Study setting').

Author's response.

Many thanks for this comment. We have now move this as suggested.

- Please add strength and limitation in the discussion section.

Author's response.

Many thanks for this comment. The strength and limitations section is on page 10 and is in a separate section from the discussion as per BMJ Open format requirements.

- The study design is a 'randomised controlled study' and page 20 (screening) describes the recruitment process as....."Recruitment will run for up to 3 years to include 3 seasonal influenza seasons between December 2017 and May 2020". Is retrospective or prospective design. Please clarify?

Author's response.

This is a prospective RCT as detailed throughout the manuscript and in the methods section. The study has been running since Dec 2017 to the present.

VERSION 2 – REVIEW

REVIEWER	Ben Cooper University of Oxford, UK
REVIEW RETURNED	21-Oct-2019
GENERAL COMMENTS	All the points raised in the previous review have now been appropriately addressed.